# PRINCIPLES-DRIVEN MACHINE LEARNING FOR UV SPECTRAL PREDICTION

## ABSTRACT

Despite recent advances in machine learning, accurate UV spectral prediction remains a challenging task. UV spectra exhibit inherently broad absorption bands characterized by peak positions, band shapes, and curvature profiles that machine learning models still struggle to capture effectively.

We present three methods that aim at identifying these three characteristics: Peak Position Awareness (PPA), Curriculum Learning for Interpolated Abstracted Spectra (CLIAS), and Spectrum Curvature Limitation (SCL).

Our performance evaluation shows that our methods can successfully capture genuine spectral characteristics, achieving consistent improvements over diverse models, especially when the training order of CLIAS and SCL is carefully considered in combination with PPA. We also show important evidence that our best models outperform the state-of-the-art UV-adVISor model.

## 1 INTRODUCTION

Advances in machine learning have achieved remarkable milestones to address intellectual tasks such as game-playing (Silver et al., 2016), multi-modal language tasks (OpenAI, 2023), and protein folding Evans et al. (2021). These successes have led to applying machine learning to scientific domains including chemistry and materials science (Choudhary et al., 2022; Keith et al., 2021).

In general, the successful development of accurate machine learning models for chemical or physical properties can lead to effective design of materials of interest. Predicting UV spectrum of an organic molecule is an important example where accurate models are necessary, due to large amounts of time and money required by experimental measurements. Accurate spectral prediction can also provide crucial insights into electronic structures and properties, with various applications including drug discovery (Barone & Polimeno, 2007) and materials science (Li & Hur, 2017; Makuła et al., 2018).

Despite numerous attempts to advance machine learning research in materials discovery (Cai et al., 2020; Liu et al., 2017), the UV spectral prediction task still remains a challenge. While training neural networks (NNs) is one way to predict UV spectra, effective training is pressed by limited data due to the difficulties and inaccuracies in both experimental measurements and theoretical calculations (Adamo & Jacquemin, 2013; Jacquemin et al., 2011) as well as limited experimental resources.

An essential challenge related to UV spectroscopy is also raised. A UV spectrum is illustrated as a curved graph where each wavelength of the ultraviolet light irradiated to a target molecule on the horizontal axis is plotted against its corresponding absorption rate on the vertical axis, and is characterized by (1) *peak positions* that are the wavelengths showing (local) maximum absorbance, (2) *band shapes* that are the wavelength ranges characterizing absorbance including intensities and full width half maximum (FWHM), and (3) *curvature profiles* that determine spectral curves (see Figure 1 for the case of rutin that is a natural compound known as flavonoid). Although these basic characteristics are established as typical ones for qualitative and quantitative molecular analysis in UV spectroscopy (Chen et al., 2024), state-of-the-art (SOTA) methods fail to capture these characteristics, suffering from accuracy and physical realism (McNaughton et al., 2023; Urbina et al., 2021).

We present three methods to capture the above characteristics: **PPA** predicts peak positions as an auxiliary classification task; **CLIAS** progressively trains on interpolated data through curriculum learning (Bengio et al., 2009) to effectively learn band shapes; and **SCL** introduces second-derivative limitation for realistic curvatures.

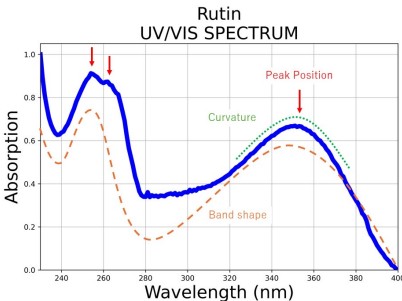

Figure 1: UV spectrum of rutin colored in blue and its characterization in UV spectroscopy

With the standard benchmark of Urbina et al. (2021), we show that a careful combination of our methods successfully improves the performance across different models. Our wavelength-specific analysis reveals that improvements achieved by our methods concentrate on less frequent yet important absorption rates. We also demonstrate predicted spectra that clearly illustrate the success of our methods and room for further improvement. Finally, we show important evidence that our best models outperform the SOTA UV-adVISor model (Urbina et al., 2021).

## 2 UV Spectral Prediction Task and Spectral Generation Model

In the setting of Urbina et al. (2021), UV range spectra are observed. For fixed range $[w, w + N - 1]$ nm of the wavelengths in the entire dataset, where $w$ and $N$ are integers, a training example for spectrum $i$ consists of a pair $(m_i, S_i)$, where $m_i$ is a molecule in SMILES format (Weininger, 1988) and $S_i$ is a spectrum to learn. $S_i$ is a sequence $y_{i,0} \to y_{i,1} \to \cdots \to y_{i,N-1}$, where $y_{i,j}$ is a real value in $[0, 1]$. Semantically, $y_{i,j}$ is an absorption rate of wavelength $(w + j)$ nm irradiated to $m_i$.

As Urbina et al. (2021) investigated, a *spectral generation model* receives input on a molecular structure and *always* yields $N$ absorption rates for corresponding $N$ wavelengths as output. The spectral generation model consists of either a multi-output regression NN model, e.g., multi-layer perceptron (MLP), or a sequence generation model, e.g., RNN. Our methods are easily incorporated into these cases to enhance the performance of the spectral generation model.

## 3 Related Work

Accurate UV spectral prediction is still difficult due to broad features including overlapping broad peaks and failing to estimate the FWHM of each peak. This section summarizes related approaches.

Time-dependent density functional theory (TD-DFT) calculations (Castro et al., 2013; Jacquemin et al., 2008) can yield theoretical UV spectra. However, they are computationally expensive and difficult to apply to large datasets and large molecular structures (McNaughton et al., 2023), resulting in calculating only limited cases such as peak positions. Additionally, the results of the calculations do not necessarily reflect the experimentally obtained UV spectra that depend on many conditions including room temperatures and solvents.

Spectroscopic knowledge is promising in other spectroscopic areas. For example, vibrational modes of McGill et al. (2021a) improved Infrared (IR) spectral prediction. Chen et al. (2023) used absorption bands for energy applications. Applying their methods to our task remains open due to different spectral characteristics.

Machine learning has been recently applied to UV spectral prediction. For example, McNaughton et al. (2023) used 3D molecular geometries and TD-DFT calculations as additional input and training data points. Their approach performed most effectively when a message passing neural network of McGill et al. (2021b) was adapted. However, their approach cannot be directly compared with our 2D SMILES-string-based approach without TD-DFT-based training data and without an extension of our methods to handle 3D coordinates.

UV-adVISor (Urbina et al., 2021) yields the current SOTA performance in UV spectral prediction. In their best-performing implementation, UV-advisor encodes a tokenized SMILES sequence to a latent vector, based on BiLSTM. The latent vector is created by a concatenation of each of hidden state in the sequence. UV-advisor also has an LSTM-based sequence-to-sequence model called the decoder that receives and generates a sequence of absorption rates. The decoder is extended with an attention mechanism based on (Luong et al., 2015), which receives the latent vector of the BiLSTM-based encoder and a hidden state of the last absorption rate in the decoder. The attention attempts to identify a relation between SMILES tokens and the last absorption. The resulting weighted vector of the attention is concatenated with the hidden state of the decoder to predict the next absorption rate. While their implementation is not publicly available, we compare the numbers in Urbina et al. (2021) with those of our best models under the same criteria, showing the superiority of our approach.

Curriculum learning (Bengio et al., 2009) progressively exposes models to increasing task complexity and has been applied to many tasks including computer vision and natural language (Wang et al., 2022) with various strategies (Soviany et al., 2022). Attempts have also been made to apply curriculum learning to scientific domains. In these attempts, physical constraints have been incorporated into various scientific domains through theory-guided data science (Karpatne et al., 2017), physical property regularization (Huynh et al., 2017), and physics-informed NNs (Chen et al., 2020). However, how to introduce such physical constraints to UV spectral prediction still remains unresolved.

Other related work includes NN-based prediction to other types of spectra namely IR spectra, e.g., (Al & Allouche, 2024; McGill et al., 2021b; Saquer et al., 2024). This task is beyond of the scope of our paper, since IR spectra exhibit much sharper peaks than UV spectra, resulting in different characteristics from the perspectives of IR and UV spectroscopy.

# 4 OUR ENHANCEMENTS TO UV SPECTRAL PREDICTION

We describe our methods motivated by the ideas behind UV spectroscopy.

## 4.1 PEAK POSITION AWARENESS (PPA)

UV spectroscopy has the fundamental principles that absorption peaks correspond to specific electronic transitions related to specific structural features within the molecule. In UV spectral analysis, as peaks are a reliable feature, chemists first identify peak positions and then consider concentration and absorption coefficients based on the Beer–Lambert law (Beer, 1852; Lambert, 1760).

PPA attempts to mimic this expert analysis process and introduces a *peak classification model (PCM)* that addresses an auxiliary task of peak classification. For wavelength range $[w, w + N - 1]$ nm, the PCM receives fixed sized input on molecular structure $m$ and predicts peak locations as a binary vector $\mathrm{pv} = (v_0, v_1, \cdots, v_{N-1})$, where $v_j \in \{0, 1\}$ $(j = 0, 1, \cdots, N - 1)$ indicates whether wavelength $(w + j)$ nm is a peak position.

PCM needs a training dataset on pv. For sufficiently large thresholds on height $h_{\mathrm{peak}}$, distance $d_{\mathrm{peak}}$ and width $w_{\mathrm{peak}}$, peak positions are computed by an algorithm to simply compare neighboring values for finding local maxima, which is available in the scikit-learn library (Pedregosa et al., 2011).

The spectral generation model is extended to receive an input concatenated $m$ with pv, and is trained with the augmented training dataset that additionally includes pv in each example, aiming at more accurate predictions with this enriched input.

For the test dataset, since peak positions are not apriori knowledge on the target molecule, the spectral generation model receives pv predicted by the PCM and generates a predicted UV spectrum.

## 4.2 CURRICULUM LEARNING FOR INTERPOLATED ABSTRACTED SPECTRA (CLIAS)

We aim at capturing band shapes of UV spectra, which provide crucial information on vibrational structure and environmental effects. Broad bands indicate extensive vibrational coupling or solvent interactions, while narrow peaks suggest rigid molecular structures with minimal vibrational freedom.

With the help of Algorithm 1, we explain CLIAS. For $[N] := \{0, 1, \cdots, N - 1\}$, CLIAS receives a subset $P \subseteq [N]$, which satisfies that (1) $P$ includes all indices for peak positions and indices 0 and

---

**Algorithm 1** CLIAS

---

**Require:** $P_1, P_2 \cdots, P_{k_a}$ where $P_1 \subset P_2 \subset \cdots \subset P_{k_a} = [N]$ for UV spectra with $N$ wavelengths
  1: INITIALIZEMODEL($\mathcal{M}$)         ▷ Random values assigned to model parameters in model $\mathcal{M}$
  2: **for** $i = 1, 2, \cdots, k_a$ **do**
  3:     $T_i = $ CREATETRAININGDATASET($P_i$) ▷ Interpolated absorption rate for wavelength $j \notin P_i$
  4:     TRAINMODEL($\mathcal{M}, T_i$)                 ▷ Training with curriculum $T_i$
  5: **end for**
  6: **return** $\mathcal{M}$

---

$N - 1$, and (2) the difference in wavelengths between two adjacent indices in $P$ are identical among all located between closest peak positions, or index 0 or $N - 1$. After training is complete with a new dataset based on $P$, CLIAS trains the model with the original dataset. As a generic form, CLIAS trains with $k_a$ subsets $P_1 \subset P_2 \subset \cdots \subset P_{k_a} = [N]$, progressing from $P_1$ toward $P_{k_a}$.

While $P$ conveys dominant, abstract shapes with $|P|$ wavelengths, $P$ is inconsistent with the model specification designed to generate $N$ absorption rates for all wavelengths. To ensure the consistency, CLIAS calculates interpolated absorption rates for the unselected wavelengths. For spectrum $i$ and two adjacent indices $j, j + L \in P$, an interpolated absorption rate for unselected wavelength $(w + j + k)$ nm (i.e. $j + k \notin P$) for abstract spectrum $i$ is defined as $((L - k)y_{i,j} + ky_{i,j+L})/L$, where $y_{i,j}$ and $y_{i,j+L}$ are the absorption rates in its corresponding original spectrum.

In practice, embodying CLIAS requires a method to construct effective subsets $P_1, P_2, \cdots, P_{k_a}$. For a preset parameter $k_a$, we currently employ a semi-automatic approach that initially includes $\lceil N/2^{k_a - i} \rceil$ indices for $P_i$ as a candidate and then manually selects a promising subset of $\{P_1, P_2, \cdots, P_{k_a - 1}\}$ as well as $P_{k_a} = [N]$.

### 4.3 SPECTRUM CURVATURE LIMITATION (SCL)

Peak curvature in UV spectra is physically constrained by natural line broadening mechanisms, e.g., lifetime broadening (Heisenberg uncertainty), Doppler broadening (thermal motion), and collisional broadening (solvent interactions). Unrealistic sharp features violate these physical constraints.

Motivated by such physical characteristics of UV spectroscopy, SCL enforces realistic curvatures by penalizing excessive curvature rarely observed in practice. For point $j$ in spectrum $i$ in the training dataset, let $d^2 y_{\mathrm{true}, i, j}$ and $d^2 y_{\mathrm{pred}, i, j}$ be respectively the true and predicted curvature values calculated as the second derivative approximation:

$$d^2 y_{i,j} \approx y_{i,j+1} - 2y_{i,j} + y_{i,j-1}. \tag{1}$$

Using standard deviation $\sigma$ of $d^2 y_{\mathrm{true}, i, j}$ for all $i$ and $j$, SCL regards predicted point $j$ in spectrum $i$ that satisfies $|d^2 y_{\mathrm{pred}, i, j}| > \sigma$ as an unrealistic one to be penalized. For realistic pairs $(i, j) \in V$ and the original loss function $\mathcal{L}(y_{\mathrm{pred}}, y_{\mathrm{true}})$, SCL defines the loss function $\mathcal{L}_{SCL}(y_{\mathrm{pred}}, y_{\mathrm{true}})$ as:

$$\mathcal{L}_{SCL}(y_{\mathrm{pred}}, y_{\mathrm{true}}) := \mathcal{L}(y_{\mathrm{pred}}, y_{\mathrm{true}}) + \lambda_{\mathrm{cur}} \sum_{(i,j) \in V} (d^2 y_{\mathrm{pred}, i, j} - b_{\mathrm{cur}})^2$$

where $\lambda_{\mathrm{cur}}$ is a hyperparameter that controls the strength of the constraint, $b_{\mathrm{cur}}$ is a hyperparameter that controls the penalty for each violated curvature, and $d^2 y_{\mathrm{pred}, i, j}$ is approximated as formula (1) to allow for training.

### 4.4 OTHER IMPLEMENTATION DETAILS

In combining PPA, CLIAS and SCL, rather than incorporating these methods at a time, we introduce a two-step training approach that performs SCL combined with PPA after no improvement is achieved by CLIAS combined with PPA (see Section 5 for detailed results).

There are several choices to encode the molecular structure, e.g., fingerprint (Morgan, 1965) and recent pretrained NNs (Gilmer et al., 2017; Honda et al., 2019). For consistency with the literature, we employ the SMILES-tokenization-based approach of Urbina et al. (2021), which partitions a SMILES string of a molecule to a sequence of tokens such as atoms and bonds. Paddings are added to

Table 1: Mean MAE values. P=PPA, C=CLIAS and S=SCL. Standard deviations are shown in the parentheses. See underline and bold numbers for the best case of each architecture and of all models.

| Model | Baseline | P | C | S | P + C | All | P+C→P+S |
|---|---|---|---|---|---|---|---|
| MLP | 0.0784 (0.0029) | 0.0681 (0.0038) | 0.0773 (0.0033) | 0.0786 (0.0029) | 0.0686 (0.0034) | 0.0679 (0.0033) | 0.0666 (0.0031) |
| LSTM | 0.0911 (0.0025) | 0.0745 (0.0032) | 0.0911 (0.0025) | 0.0910 (0.0023) | 0.0743 (0.0024) | 0.0757 (0.0031) | 0.0723 (0.0037) |
| Transformer | 0.0898 (0.0022) | 0.0746 (0.0026) | 0.0813 (0.0026) | 0.0902 (0.0020) | 0.0704 (0.0021) | 0.0694 (0.0022) | 0.0692 (0.0018) |
| BiLSTM | 0.0781 (0.0031) | 0.0672 (0.0020) | 0.0762 (0.0038) | 0.0781 (0.0027) | 0.0671 (0.0035) | 0.0681 (0.0025) | **0.0655** (0.0021) |

the end of the sequence to create a fixed-sized vector passed as input to the spectral generation model. While other effective embedding methods especially based on pretrained NNs can be combined for further enhancements, actual performance evaluation is beyond the scope of the paper.

## 5 PERFORMANCE EVALUATION

After investigating the overall performance of each of our methods as well as their synergistic impacts, we perform wavelength-specific analysis that reveals which part of the spectra our methods have successfully improved, followed by demonstrating two predicted spectra. Finally, we compare our best models against the SOTA UV-adVISor, showing the superiority of our methods.

### 5.1 SETUP

Urbina et al. (2021) prepared two small datasets obtained by physical experimental measurements. However, since larger datasets are preferred for training models in practice, we attempted to evaluate the models with the increased data size, merging these datasets into one with common wavelength ranges, except Subsection 5.5. The resulting dataset comprised 3,170 UV spectra (after removing 2 invalid SMILES) with absorption rates in the 230–400 nm range at 1 nm resolution (171 data points per spectrum), relevant to pharmaceutical development, organic electronics, and photocatalytic materials. The dataset was randomly split into training:validation:test = 7:1.5:1.5.

We aim at elucidating the behavior of our methods with diverse models. We implemented the following architectures in Python with the PyTorch library and trained them on NVIDIA H100 80GB GPUs: (1) **MLP** that generates a multi-output, (2) **LSTM** (Hochreiter & Schmidhuber, 1997) that generates a sequence, (3) **Transformer** that successfully leverages attention mechanisms (Vaswani et al., 2017), and (4) **BiLSTM** Graves & Schmidhuber (2005) that is a bidirectional version of LSTM. The PCM was implemented as an MLP.

Inspired by (Urbina et al., 2021), the MAE loss was used for training all the spectral generation models, with additional terms when SCL was incorporated. Using the peak one-hot vectors as binary labels, the binary cross entropy loss was used to train the PCM. We report the average values and standard deviations of MAE after three independent runs with random seeds (42, 123 and 456). See Appendix A for the other detailed configurations. In Subsection 5.5, we trained the models separately with the two datasets and the same evaluation metrics as (Urbina et al., 2021), allowing for fair comparisons against their numbers on UV-adVISor.

We also trained the spectral generation models with the MSE loss that is more commonly used than the MAE loss. MSE more strongly penalizes large errors, which may lead to different behaviors. However, to some extent, we observed similar trends to the results discussed in Subsections 5.2 and 5.3 (see Appendix B for details).

### 5.2 PERFORMANCE IMPROVEMENTS WITH OUR METHODS

We analyze the MAE value averaged over all wavelengths in all test spectra. This metric may underestimate infrequent yet important data, but can still capture overall performance.

Table 1 shows the performance of each model when our methods are combined. We only show the essential ones that highlight the performance of our methods rather than include all combinations. *P+C→P+S* denotes our strategy discussed in the previous section and further tunes the model with PPA and SCL with additional 30 epochs, after initially trained with PPA and CLIAS. For a comparison, we also include *All* trains a model by incorporating PPA, CLIAS and SCL all at once.

### 5.2.1 PERFORMANCE WITH SINGLE ENHANCEMENT

Starting with the case that only one of our methods is added, PPA demonstrated that peak positions provided most fundamental benefits. This significant improvement with PPA is consistent with the importance of the peaks for deeper analysis in UV spectroscopy, where peak positions are the primary principle that directly encodes the energy gap between molecular orbitals through strict quantum mechanical selection rules ($\Delta E = hc/\lambda$). On the other hand, since CLIAS and SCL are secondary and tertiary characteristics in UV spectroscopy, obtaining modest results with CLIAS and SCL is an expected behavior. In some cases, SCL worsened the performance of the baselines, indicating that SCL was not a useful method for itself. Therefore, as we discuss later, care needed to be taken when SCL was combined with PPA and CLIAS.

PPA assumes the high performance of the PCM. Our PCM achieved the precision of 0.868 and the recall of 0.908. The average error of the wavelength for the peak position was 1.55 nm. These numbers were sufficient to achieve significant improvement with PPA.

Baseline Transformer was not sufficient to accurately predict UV spectra, underperforming baselines MLP and BiLSTM. In natural language processing, attention plays a crucial role in Transformer achieving high performance Vaswani et al. (2017). However, in predicting UV spectra, since Transformer receives the identical, repeated input on molecular structure, the positional encoder is the only factor that can differentiate the keys, values and queries of its self-attentions. This could be one reason Transformer failed to capture the relations between different wavelengths effectively.

### 5.2.2 COMBINATION OF PPA WITH CLIAS

PPA with CLIAS (P+C) tended to slightly improve PPA roughly by 5.6% in the best case, which was still essential for more accurate spectral prediction.

CLIAS focuses on band shapes that are a secondary feature and less crucial than peak positions. The modest improvement with P+C consistently mirrors the physical reality of UV spectroscopy, e.g., vibrational structure and solvent effects through the Franck-Condon principle (Atkins & Friedman, 2011; Condon, 1926). It is after the primary feature of peak positions is established that the Franck-Condon principle governs intensity distribution through vibrational progressions, and band shapes encode essential information about molecular geometry changes during electronic transitions. In a sense, CLIAS is regarded as an attempt to capture these envelopes through progressive resolution refinement, learning the flows on spectral intensity from the peak positions defined by PPA.

### 5.2.3 COMBINATION OF PPA WITH CLIAS AND SCL

PPA with CLIAS and then PPA with SCL (P+C→P+S) allowed P+C to first establish a stable solution for peak positions and an overall shape and provided P+S room with a better initialization point for curvature refinement. In the best case, P+C→P+S performed 4.6% better than the model only with P+C. Compared with the baselines, the final P+C→P+S models achieved significant improvements, i.e., 15.1% for MLP, 20.7% for LSTM, 23.0% for Transformer, and 16.2% for BiLSTM, respectively.

We note that P+C→P+S is consistent with UV spectroscopic analysis based on natural line broadening mechanisms, considered as a tertiary approach after peaks and band shapes are established. Given peak positions and band envelopes, natural line broadening mechanisms determine the final spectral smoothness. SCL is regarded as an embodiment to enforce these practical, physical constraints to prevent unrealistic sharp features while preserving the essential spectral structure.

On the other hand, for the *All* strategy that combines all three methods at one time, there were both cases where the performance was improved or degraded compared to P+C. These mixed results are attributed to the fact that the capability of SCL penalizing sharp shapes cannot effectively work when basic spectral patterns have not been learned yet, such as in an early training phase.

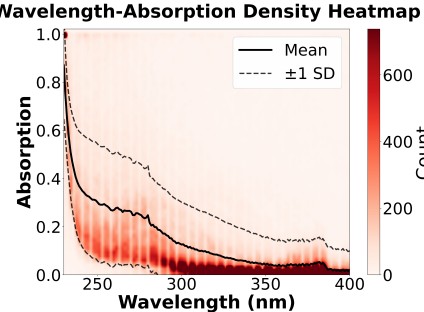

Figure 2: Absorption rate heatmap, mean absorption rate (solid line) and standard deviation (dashed line) for each wavelength in the training dataset

### 5.3 WAVELENGTH-SPECIFIC PERFORMANCE ANALYSIS

We aim at further identifying the advantages of our methods, which were failed to capture by the mean MAE value discussed in the previous subsection. Our analysis for representative wavelengths elucidate that our methods successfully improve predictions to outlier-like absorption rates observed in spectra, which consist of a much smaller number of examples.

#### 5.3.1 DISTRIBUTION OF ABSORPTION RATE FOR EACH WAVELENGTH

We start by analyzing the spectra in the training dataset. Figure 2 illustrates the statistics on the mean absorption rate and standard deviation of each wavelength for the spectra in the training dataset. The mean absorption rate exhibits high values ($> 0.6$) at 230–250 nm, gradually decreasing to near-zero ($< 0.1$) at 350–400 nm. The wider statistical variance at short wavelengths reveals the coexistence of both strong and weak absorbers that yield high and low absorption rates, while all molecules show negligible absorption at longer wavelengths as evidenced by the smaller statistical variance.

This distribution reflects the physical reality of UV spectroscopy: Most organic molecules absorb strongly in the deep UV (200–280 nm) due to $\pi \rightarrow \pi^*$ transitions, while absorption in the near-UV (350–400 nm) requires extended conjugation systems that occur relatively rarely.

#### 5.3.2 MODEL PERFORMANCE FOR REPRESENTATIVE WAVELENGTHS

On the basis of the spectral patterns shown in Figure 2, we selected four representative wavelengths (230, 250, 300, and 350 nm) and analyzed the BiLSTM-based models with the test dataset.

Figure 3 shows the histogram of the absorption rates and the average MAE values of the baseline, P+C and P+C→P+S for each categorized absorption rate. The histogram on the test dataset mirrors the patterns revealed with the training dataset. Of the 476 spectra in the test dataset, over 300 spectra had absorption rates of $> 0.9$ for 230 nm. Less lopsided peaks were shown for 250 nm, and over 400 spectra exhibited near zero values clustered in the lowest absorption bin for $\geq 300$ nm.

Thus, UV spectral prediction faces a challenge due to wavelength-dependent asymmetric distributions where frequencies of absorption values may differ over 100 times. This challenge was hidden in the previous subsection, due to the assessment with the MAE value averaged by popular absorptions.

Irrespective of the wavelengths, when irradiated molecules yielded frequent absorption rates with those wavelengths, all models predicted those absorption rates accurately. There was only small room for further improvement in this case. On the other hand, both P+C and P+C→P+S successfully improved the baseline for infrequent absorption rates. P+C→P+S tended to generate even better MAE values than P+C. From a UV spectroscopic perspective, addressing these infrequent absorptions is also of importance, since genuine understanding of spectral physics is often required. Improved MAE values show that our methods captured such genuine characteristics of the UV spectra better.

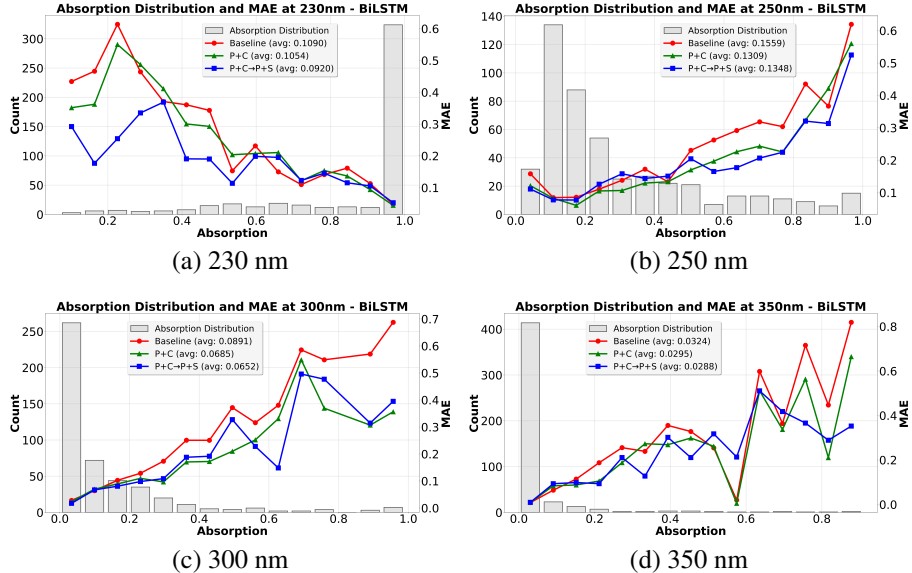

Figure 3: Comparisons across four wavelengths. Histograms show the counts of the spectra, categorized by the absorption rates yielded for the wavelengths in the captions. Overlaid lines indicate MAE values of baseline (red), P+C (green) and P+C→P+S (blue) for each absorption rate bin.

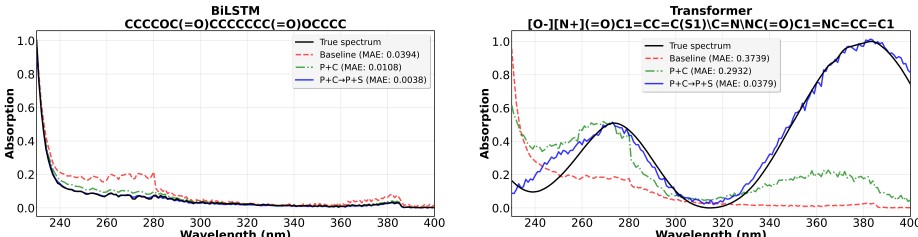

Figure 4: Spectra generated by baseline (red), P+C (green), and P+C→P+S (blue) and compared against ground truth (black)

### 5.4 ANALYSIS ON GENERATED UV SPECTRA

Figure 4 illustrates two spectra generated by our models, showcasing that our methods significantly improved the baseline models toward ground truth spectra from a viewpoint of realistic shapes.

For the BiLSTM spectrum (left), the baseline model strongly predicted peaks around 280 nm and 380 nm that were absent in the true spectrum. The P+C method (green line) began to correct this by capturing a more appropriate spectral shape, reducing the MAE to 0.0189. Furthermore, P+C→P+S (blue line) achieved remarkable improvement, closely approximating the true spectrum with an MAE of only 0.0032. This progression improvement demonstrates how our methods systematically refined predictions from incorrect peak assignments to accurate spectral reproduction.

For the Transformer spectrum (right), while all models had difficulties in generating accurate shapes, our methods still worked more effectively than the baseline. The baseline prediction was almost entirely different from the true spectrum, as evidenced by its high MAE of 0.3739. Its strong peak at 230 nm appeared to be affected by other training examples. With P+C, the model began to identify the broad peak positions around 270 nm and 380 nm, although they fit only partly with an MAE of 0.2932. P+C→P+S significantly reduced the MAE to 0.0379 with better fit particularly around 380 nm. However, the smooth nature of the true spectrum was not fully reproduced, with some roughness remaining in the predicted curve. This suggests that while our methods successfully learned peak positions, additional refinement may be needed for capturing smooth spectral profiles.

Table 2: Comparisons with *test* datasets (median MAE and standard deviations in parentheses). The best numbers in Tables S2 and S3 of Urbina et al. (2021) are extracted for UV-adVISor. Bold fonts are the cases when the "best" P+C→P+S models are selected by their *validation* dataset performance.

| Model | UV-adV. | Baseline | | | | P+C→P+S | | | |
|---|---|---|---|---|---|---|---|---|---|
| | | MLP | LSTM | Trans. | BiLSTM | MLP | LSTM | Trans. | BiLSTM |
| Dataset I | 0.091 (0.12) | 0.0752 (0.0044) | 0.0905 (0.0024) | 0.0846 (0.0025) | 0.0760 (0.0072) | **0.0518** (0.0013) | 0.0626 (0.0033) | 0.0568 (0.0023) | 0.0579 (0.0044) |
| Dataset II | 0.044 (0.052) | 0.0470 (0.0025) | 0.0450 (0.0003) | 0.0453 (0.0005) | 0.0510 (0.0029) | 0.0410 (0.0016) | 0.0400 (0.0010) | **0.0403** (0.0013) | 0.0408 (0.0021) |

## 5.5 COMPARISON WITH STATE-OF-THE-ART UV-ADVISOR

We attempt to answer how competitive our models are with the SOTA under the same training and evaluation criteria as (Urbina et al., 2021): After training our baseline and P+C→P+S models on datasets I and II, respectively, we evaluated them on the test datasets with median MAE.

Table 2 compares the performance of our models against the SOTA UV-adVISor, where the median MAE was calculated as an average value of three runs. For both datasets, our baselines already tended to be competitive to UV-advisor, and all of our P+C→P+S models outperformed UV-adVISor. From a practical viewpoint, the models performing best for the test datasets may not always be able to be selected due to the unavailability of the true absorption rates. Our results, therefore, ensure the superiority of our methods even if the final P+C→P+S models to work on the test datasets need to be selected by their best performance on the validation datasets. The performance difference was more clearly observed for dataset I that was more difficult for all models. P+C→P+S MLP yielded the MAE value that was 40% better than that of UV-adVISor.

From an algorithmic viewpoint, UV-advisor leverages the molecular structure indirectly: It attempts to identify important atoms and bonds that contribute to absorption via attention calculated by the concatenated hidden states of the tokenized SMILES sequence and the hidden state of the last absorption rate, which is then represented as a weighted vector. With the hidden state of the absorption rate, this weighted vector is used as an important feature to predict the next absorption rate.

In contrast, our baseline models leverage the molecular structure (i.e., tokenized SMILES sequence) as a more direct feature received as an input to the spectral generation model. Our results indicate that such a molecular structural feature was sufficient to create the competitive models. Further enhancing these baselines with our methods was crucial to outperform UV-adVISor.

## 6 CONCLUSIONS AND FUTURE WORK

We introduced PPA, CLIAS and SCL, which embody fundamental physical principles and analytical techniques behind UV spectroscopy. With the dataset obtained by the physical experiments (Urbina et al., 2021), we showed that P+C→ P+S achieved at least 15% improvement over the diverse baseline models. Our wavelength-specific analysis revealed that our methods improved the performance for infrequent but important absorption rates in the asymmetric distributions. These advantages have led to generating more realistic UV spectra than the baselines. Our performance evaluation finally showed strong evidence that our best models outperformed the SOTA UV-adVISor.

There are various approaches to explore as future work. For example, imbalanced learning techniques, e.g., (He & Garcia, 2009; Chawla et al., 2002) have a potential to resolve issues caused by the asymmetric distributions on the absorption rates. Extending the dataset size is clearly of importance, and opens up opportunities to develop better theoretical and experimental methods in UV spectroscopy, including a combination of TD-DFT and experimental data. New machine learning algorithms that leverage partially observed spectra such as from the NIST database[1] also help effectively train models with extended data. Finally, extending our methods to address other spectroscopic tasks that have different characteristics is important, e.g., IR spectroscopy with sharp vibrational peaks, NMR with discrete chemical shifts, and mass spectrometry with fragmentation patterns.

---

[1]https://webbook.nist.gov/chemistry/

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

Table 3: Mean RMSE values. P=PPA, C=CLIAS and S=SCL. Standard deviations are shown in the parentheses. See underline and bold numbers for the best case of each architecture and of all models.

| Model | Baseline | P | C | S | P + C | All | P+C→P+S |
|---|---|---|---|---|---|---|---|
| MLP | 0.1213 (0.0038) | 0.1087 (0.0069) | 0.1206 (0.0037) | 0.1233 (0.0036) | 0.1057 (0.0017) | 0.1049 (0.0048) | 0.1010 (0.0041) |
| LSTM | 0.1436 (0.0025) | 0.1184 (0.0029) | 0.1432 (0.0024) | 0.1434 (0.0025) | 0.1184 (0.0052) | 0.1170 (0.0021) | 0.1140 (0.0048) |
| Transformer | 0.1416 (0.0033) | 0.1160 (0.0023) | 0.1259 (0.0029) | 0.1402 (0.0030) | 0.1098 (0.0035) | 0.1100 (0.0027) | 0.1086 (0.0036) |
| BiLSTM | 0.1175 (0.0063) | 0.1081 (0.0089) | 0.1198 (0.0059) | 0.1226 (0.0066) | 0.1038 (0.0047) | 0.1061 (0.0066) | **0.1003** (0.0036) |

## A   DETAILED SETUPS FOR PERFORMANCE EVALUATION

The model architectures of our baseline models are summarized as follows:

- **MLP**: Four hidden layers with sizes of 1024, 512, 256, and 128 units respectively, followed by 171 output units, using ReLU activation (Nair & Hinton, 2010)

- **LSTM**: Single-layer LSTM with 256 hidden units followed by a linear layer to 171 outputs

- **BiLSTM**: Three-layer bidirectional LSTM with 512 hidden units per direction, followed by three fully connected layers of sizes 1024, 512, and 256 units with ReLU activation, and finally 171 outputs

- **Transformer**: Four encoder layers with 8 attention heads, model dimension of 256, and feedforward dimension of 1024, followed by two fully connected layers with 256 units each using ReLU activation, and finally 171 outputs

The peak prediction model consists of a 3-layer MLP with each hidden layer size 2048, 1024, 512, and 256, followed by 171 outputs with a sigmoid operation to calculate a probability that each absorption rate is a peak.

A molecule is encoded as SMILES string at most with 150 characters, tokenized by common chemical vocabularies. The padded tokens are embedded to a vector of size 256, which is then passed to the input of each NN. When PPA is combined, another embedded vector of size 256 is created from the peak-value vector of size 171, concatenated with the embedded vector on the molecule.

We performed extensive hyperparameter optimization using the validation set. For peak detection, we found optimized parameters through grid search: $h_{peak} = 0.1$ (height threshold), $d_{peak} = 25$ nm (minimum distance), and $w_{peak} = 20$ (peak width). The optimized prediction threshold for the peak classification model was set to 0.3. For SCL, we set $\lambda_{cur} = 0.1$ and $b_{cur} = 0.1$.

In model training, we set early stopping with a patience of 30 epochs and the Adam optimizer (Kingma & Ba, 2015) with optimized learning rates: $10^{-3}$ for MLP, $10^{-4}$ for Transformer, and $5.0 \times 10^{-4}$ for the others. Dropout (rate=0.2) (Srivastava et al., 2014) was applied between fully connected layers.

For CLIAS, in our preliminary evaluation, we set $k_a$ to at most 3 and tested three combinations of abstraction sizes: [43,171], [86,171], and [43,86,171] The best-performing configurations were [43, 86, 171] for MLP and LSTM, and [43, 171] for BiLSTM and Transformer.

Each phase except the final phase used early stopping (patience=20) and learning rate decay of 0.8 between phases, while the final phase used patience=7.

## B   PERFORMANCE EVALUATION BASED ON RMSE

We show the performance when we trained the models under the same conditions except that the MSE loss was used, and evaluated them on the test dataset with the RMSE values.

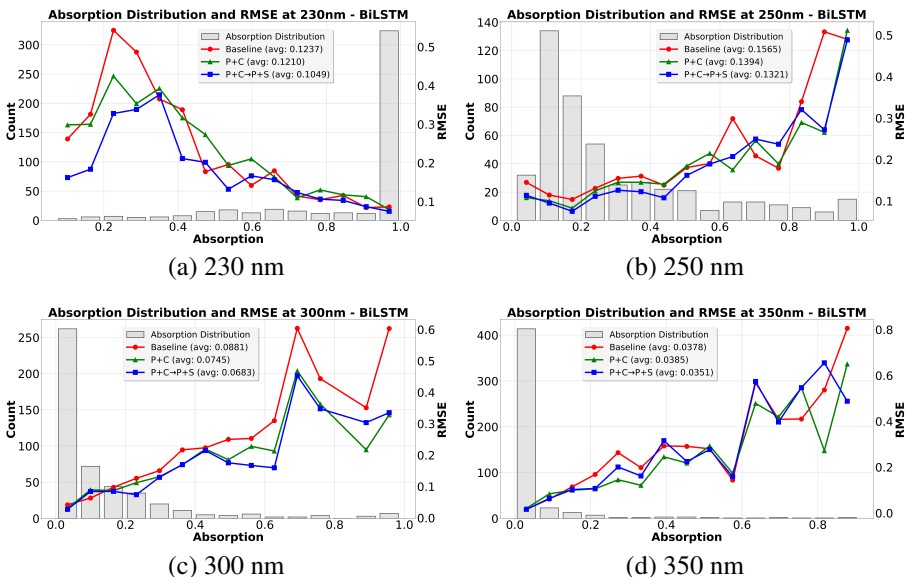

(a) 230 nm  (b) 250 nm

(c) 300 nm  (d) 350 nm

Figure 5: Comparisons across four wavelengths. Histograms show the counts of the spectra, categorized by the absorption rates yielded for the wavelengths in the captions. Overlaid lines indicate RMSE values of baseline (red), P+C (green) and P+C→P+S (blue) for each absorption rate bin.

**Overall Performance Improvements**   Table 3 shows the performance of each model when our methods are combined. We observed the performance improvements that were similar to those with MAE. For a single enhancement, PPA achieved the most significant improvement as a primary UV spectroscopic principle, followed by CLIAS as a secondary one, while there were cases where SCL underperformed the baselines. As a combined enhancement, P+C improved PPA roughly at most by 5.4%, while 5.5% was the best improvement of P+C→P+S over P+C. Compared to P+C→P+S, *All* did not work effectively. Compared to the baselines, the final P+C→P+S models achieved significant improvements, i.e., 16.6% for MLP, 20.6% for LSTM, 23.4% for Transformer, and 14.6% for BiLSTM, respectively.

**Wavelength-specific Analysis**   Figure 5 shows the histogram of the absorption rates and the average RMSE values of the baseline, P+C and P+C→P+S for each categorized absorption rate. In general, as observed in Subsection 5.3, all models tended to accurately predict the absorption rates that appeared frequently, while both of P+C and P+C→P+S successfully improved the baseline for infrequent absorption rates. On the other hand, while P+C→P+S tended to perform better than P+C, it slightly became harder to capture this trend with the graphs shown here.

