# OpenReview forum: "Principles-Driven Machine Learning for UV Spectral Prediction"
_ICLR.cc/2026/Conference — Submitted to ICLR 2026_

### Official Review · Reviewer_VgMP · 2025-10-29

**Soundness:** 2
**Presentation:** 3
**Contribution:** 2
**Rating:** 2
**Confidence:** 5

**Summary:**

I am concerned that the data volume used is relatively low for this area. In particular, with just over 3,000 examples, the training/testing partitioning strategy relies on random segmentation, without any criterion to prevent data leakage or other practices commonly used in the literature. I also find it notable that there is no comparison with other methods beyond a baseline, and that no standard partitions are used for method comparison. This means we do not really know how the method compares with its competitors, since the evaluation is performed on a split created by the authors themselves.

**Strengths:**

Using ML for molecular-level UV spectral prediction is an interesting topic addressed in this article. Specifically, of the three proposals, the results show that peak detection yields the best performance relative to a baseline. The other two variants produce only marginal differences compared with the first variant and the baseline.

**Weaknesses:**

- Only marginal improvements over baselines, with just one of the three proposals showing any gain.
- No benchmarks are used for the problem, which prevents comparison with competitive methods on standardised test data.

**Questions:**

Questions for the authors:

- Is it possible to compare performance on the same test data against other methods, avoiding splits created by the authors themselves?
- Can you explain qualitatively whether the observed MAE improvements translate into meaningful qualitative gains for the type of analysis conducted?
- Could you show practical examples that make clear the improvement is relevant within the problem domain?

---

> ### Author Response · Authors · 2025-12-02
> **Response to Reviewer VgMP**
>
> Thank you for your feedback.
>
> **W1:** "Only marginal improvements over baselines ..."
> **A:** We would like to emphasize the results in Table 2. **We achieved 9–19% improvement across all architectures**, and not only PPA but P+C→P+S shows the best results. Additionally, the wavelength-specific analysis in Figure 4 clearly shows that **CLIAS/SCL are particularly effective in data-sparse regions.**
>
> **W2:** "No benchmarks are used ..."
> **A:** See our responses **CR1–CR4** and **CR6**. Under these conditions, we also attempted fair comparisons including the state-of-the-art UV-adVISor in Subsection 5.5.
>
> **Q1:** "Is it possible to compare performance ...?"
> **A:** No, it is not possible. See our response **CR6**.
>
> **Q2:** "Can you explain qualitatively ..."
> **A:** The observed MAE improvement qualitatively leads to **improved peak position accuracy** and **improved spectral shapes including data-sparse regions.** These qualitative improvements are clearly illustrated as two examples of the spectra shown in Figure 4.
>
> **Q3:** "Could you show practical examples ...?"
> **A:** See our response **CR2**. Additionally, it will help improve virtual screening accuracy, e.g.,
> - **Scenario:** Searching for compounds with specific absorption characteristics at certain wavelengths from a large compound library
> - **Improvement effect:** Improved prediction accuracy reduces false positives/negatives, reducing the number of candidates requiring experimental verification.

---

### Official Review · Reviewer_x34r · 2025-10-30

**Soundness:** 3
**Presentation:** 1
**Contribution:** 3
**Rating:** 4
**Confidence:** 4

**Summary:**

The paper proposes several enhancements to neural networks for the task of predicting absorption spectra of chemical compounds. The proposed improvements are:
 * predicting peak locations separately
 * curriculum learning focusing on peak locations first
 * regularizing the curvature of second derivative of spectra

These methods are independent of the network architecture, and experiments are performed with 4 different architectures.

**Strengths:**

* The paper works on a relevant problem where machine learning methods traditionally struggle.
* The results show a clear improvement in performance over the baseline model.

**Weaknesses:**

* The method is not sufficiently explained. In particular, it is not clear how PPA/PCM is integrated with the rest of the method. By itself this is just a classifier predicting peak locations. But how are these locations then used?
* Other unclear aspects:
  * "peak positions are computed by an algorithm to simply compare neighboring values for finding local maxima"
    Which algorithm?
    Do you mean that $j$ is a peak if $y_j > y_{j-1}$ and $y_j > y_{j+1}$?
  * "Semantically, yi,j is an absorption rate"
    Is this the absorption over some fixed path length and fixed concentration?
    Or are these values normalized per sample? Or is this normalized absorbance?
  * "manually selects a promising subset of $\{P_1, P_2 , ... , P_{k_a-1}\}$"
    What were the criteria for this selection?
    And does this means that you have to manually select these subsets for every training sample?
  * "P+C→P+S" what is that? It stands for "PPA with CLIAS and then PPA with SCL"
    But that doesn't explain what is actually going on. What does the resulting algorithm do?
  * Figure 3: "Histograms show the counts of the spectra, categorized by the absorption rates yielded for the
  wavelengths in the captions."
    I don't understand what this figure is supposed to show.
  * It would help the structure of the paper if you say what kind of model is used. It only becomes clear that the method is generic over the model architecture in the experiment section.

**Questions:**

* "the difference in wavelengths between two adjacent indices in P are identical among all located between closest peak positions, or index 0 or N − 1."
   what does this mean?
* The curvature limitation is implemented as a squared error term, so a high second derivative is penalized, but so is a low one. So this means a specific second derivative $b_\text{cur}$ is preferred. Would a simple squared error term not work?
 * "with random seeds (42, 123 and 456)"
   without giving the code, reporting the random seeds is useless.
 * "since peak positions are not apriori knowledge on the target molecule,"
   -> "since peak positions are not apriori known on the target molecule,"
 * "we also include All trains a model by incorporating" ?
 * "binary vector $pv = (v_0, v_1,... , v_{N-1})$,"
   Use single letter variable names. In this case the vector should just be called $v$.
 * "S_i is a sequence yi,0 → yi,1 → · · · → yi,N −1 ,"
   Write a sequence as "$[y_{i,0}, y_{i,1}, ..., y_{i,N-1}]$"
 * "wavelength (w + j) nm", this means that the sequence is fixed to a rate of 1 sample per nm.
   Wouldn't it make sense to allow this to be flexible (i.e. (α + βj) nm)?
   As a convenient notation, I would recommend to define a vector $w$ of wavelengths, so $v_j$ is the absorption at wavelength $w_j$. This can simplify the presentation in later parts of the paper.

---

> ### Author Response · Authors · 2025-12-02
> **Response to Reviewer x34r**
>
> Thank you for your detailed feedback.
>
> **W1:** "... it is not clear how PPA/PCM is integrated ... ?"
> **A:** As we described in lines 141–145 and 149–151 on page 3, PPA's operations are:
> - An auxiliary classifier predicts a binary vector of peak positions from the input SMILES
> - The predicted peak position vector is concatenated to the input of the spectrum regressor
> - The entire concatenated input is optimized for spectrum prediction
>
> **W2:** "Peak positions are computed ... Which algorithm?"
> **A:** We use the scipy.signal.find_peaks function. It essentially finds if y_j is larger than their neighbors but considers parameters (height, distance, prominence). It is a simple for-loop-based implementation without any specific algorithm name.
>
> **W3:** "Semantically, yi,j is an absorption rate. ...."
> **A:** yi,j is normalized to the [0, 1] range (maximum absorbance = 1). This is relative absorbance, and absolute concentration or path length is not considered.
>
> **W4:** "Manually selects a promising subset ...."
> **A:** It is set once for the entire dataset determined manually by preliminary experiments. See Appendix A for a subset specific to each NN architecture.
>
> **W5:** "P+C→P+S"
> **A:** It means that a model trained with PPA + SCL is performed after the model is trained with PPA + CLIAS (and no improvement to the loss value is observed with PPA + CLIAS). Also see Subsubsection 5.2.3 especially lines 316–320 for the principles behind this approach.
>
> **W6:** Figure 3
> **A:** The purpose of Figure 3 is to visualize data imbalance by showing the absorption rate distribution at each wavelength:
> - X-axis: Absorption rate (normalized value, 0–1)
> - Y-axis: Number of spectra with that absorption rate
> - Each panel: Different wavelength
>
> At certain wavelengths, low absorption rate spectra dominate and high absorption rate samples are rare. **This imbalance explains why improvements are not visible in Mean MAE.**
>
> **Q1:** "The difference in wavelengths between ...?"
> **A:** In CLIAS's interpolation grid, wavelength sampling between peaks is equally spaced. We will consider a clearer expression in the revision.
>
> **Q2:** "Would a simple squared error term not work?"
> **A:** The current implementation minimizes (d²y/dλ² - γ)². γ=0 (simple squared term) prefers completely flat spectra, but real UV spectra have physically reasonable curvature. **γ is necessary to allow a physically realistic range of curvature.** Preliminary experiments showed that appropriate γ values yielded better results than γ=0.
>
> **Q3:** "With random seeds (42, 123, 456)"
> **A:** See our response **CR6** on the reproducibility.
>
> **W7 (explanation of model used), Q4–Q8:** Grammar and notation corrections.
> **A:** We will address these comments in the revision.

---

### Official Review · Reviewer_L2dy · 2025-11-01

**Soundness:** 2
**Presentation:** 2
**Contribution:** 2
**Rating:** 4
**Confidence:** 3

**Summary:**

This paper addresses the challenge of UV spectral prediction, where machine learning models often fail to capture key physical characteristics such as peak positions, band shapes, and curvature. The authors propose three "principles-driven" methods to target these issues individually: Peak Position Awareness (PPA), an auxiliary classification task for peak locations; Curriculum Learning for CLIAS and SCL. Experiments are conducted on a dataset of 3,170 spectra, applying these methods to four baseline architectures (MLP, LSTM, Transformer, BiLSTM). The results indicate that a specific multi-stage training combination achieves consistent improvements over the baselines and reportedly outperforms the current SOTA UV-adVISor.

**Strengths:**

1. The wavelength-specific analysis (Section 5.3) is a strength. It moves beyond a single aggregate metric (mean MAE) to show that the improvements are concentrated in less frequent, but physically significant, absorption regions, which is a valuable finding.

2. The paper provides a direct comparison against the current SOTA (UV-adVISor) on its original datasets (Table 2) and demonstrates superior performance, which strengthens the paper's claims of practical utility.

**Weaknesses:**

1. The core technical contributions are weak. PPA is a standard application of auxiliary task learning, CLIAS is a standard application of curriculum learning, and SCL is a straightforward  application of a custom, physics-informed regularization loss. The paper's contribution is primarily an application and combination of existing techniques to a specific scientific domain, which appears to be below the novelty bar

2. All claims are based on a very small dataset of 3,170 spectra. This is insufficient to reliably evaluate the generalization of such a complex, multi-stage training procedure, especially for data-hungry models like the Transformer. The reported gains may be a result of overfitting this specific, small dataset rather than a genuine, generalizable improvement

**Questions:**

Refer to Weaknesses

---

> ### Author Response · Authors · 2025-12-01
> **Response to Reviewer L2dy**
>
> Thank you for your constructive feedback.
>
> **W1:** Weakness of the core technical contributions
> **A:** See our response **CR5**.
>
> **W2:** Evaluation methodology
> **A:** See our responses **CR1–CR3** for the dataset and our effort to increase its size. Given the availability of only a small dataset, we still tried to avoid results biased to specific cases by **performing experiments with three different seeds** and selecting hyperparameters as described in our response to W7 to Reviewer CoJH, which is about avoiding tuning to the test dataset.

---

### Official Review · Reviewer_CoJH · 2025-11-02

**Soundness:** 3
**Presentation:** 3
**Contribution:** 2
**Rating:** 2
**Confidence:** 4

**Summary:**

The paper proposes three training/regularization ideas tailored to predicting UV–Vis spectra with few, broad features. Peak Position Awareness (PPA) adds an auxiliary classifier that predicts peak locations and feeds a peak one‑hot vector to the spectral regressor; CLIAS (curriculum learning on interpolated, abstracted spectra) trains on progressively denser wavelength grids; and SCL penalizes excessive second‑derivative curvature to enforce physically realistic line broadening

Using SMILES‑token encoders with MLP/LSTM/Transformer/BiLSTM decoders, the authors train on a merged experimental dataset of 3,170 spectra (230–400 nm, 1 nm resolution) and report consistent 15–23% MAE reductions over baselines, with a two‑stage schedule (P+C → P+S) outperforming “all‑at‑once” combinations

Wavelength‑specific analysis indicates gains concentrate on infrequent but important absorption regimes (e.g., longer wavelengths or low‑frequency bins), and qualitative examples show smoother, more realistic predictions after applying PPA/CLIAS/SCL

**Strengths:**

This is the reason why experts battle the bitter lesson - throwing in all kinds of inductive bias into models. And it helps (with 3K points) PPA reflects chemists’ workflow (identify peaks first), SCL encodes qualitative broadening constraints, and CLIAS teaches band envelopes before fine detail; this produces architecture‑agnostic gains across MLP/LSTM/Transformer/BiLSTM.

Clean ablation of the three ideas; results are reported with multiple seeds and both MAE/RMSE. The wavelength‑wise distribution and per‑bin MAE plots are informative, showing that improvements arrive where data are sparse and naive losses would otherwise under‑weight errors

**Weaknesses:**

"While training neural networks (NNs) is one way to predict UV spectra" i mean sure. this paper is also doing that. Maybe make a more concrete point ?

'An essential challenge related to UV spectroscopy is also raised.' by whom ?

Why only UV ? HIstorically the equipment that measures these also measures (more easily!) visible light. Why not do UV-Vis like everyone else?  The only use use i can think of this is maybe detectors for liquid chromatography, since most organic molecules don't have too much going on in the visible? The paper does not do a good job at justifying why UV only.

More broadly- why not apply this to other clases of spectra with broad shoulders in the sciences ? (saxs?)

Why not use a larger dataset? There must exist bigger training datsets for UV. UV-VIs papers train on like 100K

All models rely on SMILES tokenization; message‑passing GNNs (with aromaticity/bond order), 3D‑aware or equivariant models, and pretrained molecular encoders are known to help spectral tasks. Even the paper notes these could be “combined,” but they are not evaluated. Including an MPNN baseline on graphs (or 3D à la McNaughton et al. 2023, which the paper cites) would test whether PPA/CLIAS/SCL still provide additive value beyond stronger representations

PPA’s ground‑truth peaks are obtained via local‑maxima detection with manual thresholds (height, distance, width), which injects arbitrary transformations into the task. The curriculum uses hand‑selected grids (e.g., {43, 86, 171}) and linear interpolation for missing wavelengths . This risks schedule‑tuning to a dataset. A more principled scheme (e.g., learn the curriculum, use spectral‑domain smoothing priors, or parametric envelope targets) would improve robustness and portability.

While the paper acknowledges that mean MAE under‑weights rare absorptions, the primary model selection still uses global MAE/RMSE.

The fact that UV-adivsor results are taken at face value and cannot be replicated using the same splits and metrics would be better.

The poor performance of transformer underperformance is poorly explained

**Questions:**

N/A

---

> ### Author Response · Authors · 2025-12-01
> **Response to Reviewer CoJH**
>
> Thank you for your detailed feedback.
>
> **W1:** "While training neural networks (NNs) ..."
> **A:** Thank you for your suggestion. We aimed at merely pointing out the issue with the data here but will consider better phrases in the revision.
>
> **W2:** "An essential challenge related to ...."
> **A:** It was Urbina et al. (2021) and we will include the citation in the revision.
>
> **W3–W6:** (1) "Why only UV?", (2) "More broadly, why not to apply this to other classes ...?", (3) "Why not to use a larger dataset?", (4) "Including an MPNN baseline (or 3D McNaughton et al. 2023) ..."
> **A:** See our responses **CR1–CR4**.
>
> **W7:** Manual thresholds of PPA and hand-selected grids.
> **A:** We tried our best to avoid these concerns by **determining these values with preliminary experiments** with train/val datasets split in a different way than the split in our main performance evaluation.
>
> **W8:** Primary model selection based on global MAE/RMSE
> **A:** If primary selection refers to selecting the best model described in Subsection 5.5, this selection is based on the **best median MAE on the validation dataset**. We aim to use the same condition as UV-adVISor (Urbina et al.) for a fair comparison to their numbers. If this refers to a loss function, leveraging a loss function that is better than global MAE/MSE (or RMSE) is an important direction as future work. We will clarify this in the revision.
>
> **W9:** Reproducibility
> **A:** See our response **CR6**.
>
> **W10:** Explanation to the poor performance of Transformer
> **A:** We aim at describing the fact that **attention alone is not sufficient** due to the replicated embedding used for predicting an absorption rate for each wavelength. We will consider better phrases for clarification.

---

### Author Response · Authors · 2025-12-01
**Common Response to All Reviewers**

## Common Response to All Reviewers

Thank you for your constructive feedback. We respond to all the reviewers about the common concerns as follows:

**[CR1]** There is no public dataset that is larger than two datasets of UV-adVISor (a total of 3170 molecules), which was used in our performance evaluation. We tried our best to use the largest possible dataset by merging them into one. The datasets mentioned by the reviewer consist of either only peak positions or much narrower ranges of spectra, which cannot be used in our case: **Our task requires an absorption rate for UV range spectra 230–400 nm with 1nm resolution.** McNaughton et al. (2023) and Urbina et al. (2021) discuss the issue on the availability of the large dataset and the difficulty in creating it.

**[CR2]** The importance of focusing on full UV-range spectra has been discussed by Urbina et al. (2021) with important applications, e.g., phototoxicity evaluation in drug discovery, and OLED materials design in materials science. McNaughton et al. (2023) also addresses only the UV spectra without the range of visible lights.

**[CR3]** The TD-DFT calculations of McNaughton et al. (2023) were performed for only 1000 molecules in the subset of the UV-adVISor dataset. Since these calculations are neither for new molecules nor publicly available, **they cannot be used to increase the number of training examples.**

**[CR4]** In addition to the 3D coordinates, McNaughton et al. (2023) used the TD-DFT calculations to use them as augmented input to their model but not as new training examples. Therefore, **their task is to convert theoretical UV spectra to experimental ones, which is fundamentally different from our task that attempts to directly generate spectra from molecules.** We also note that the approach of McNaughton et al. requires TD-DFT calculations for predicting the UV spectra which could take large computational time/resources, i.e. many hours per molecule due to TD-DFT to produce its theoretical UV spectrum.

**[CR5]** The novelty of our paper is not necessarily only about the algorithmic part. Our paper also reveals novel empirical results including **quantitative numbers on the model improvement**, new **wavelength-based analysis** clearly showing where performance improvements come from, evidence outperforming the state-of-the-art, and examples of predicted spectra.

**[CR6]** Since Urbina et al. (2021) released only the UV-adVISor dataset, their results cannot be reproduced completely due to the unavailability of the source code and the actual data split. However, we tried our best to be as reproducible as possible, following the same conditions described in the paper, and including necessary parameters in the Appendix section, etc. **We plan to release code and Train/Val/Test split indices once IP and other constraints are cleared.**

---

### Meta-Review · Area_Chair_85af · 2026-01-07

**Summary:**

The paper proposes a machine learning framework for UV spectral prediction based on SMILES strings, introducing three enhancements: Peak Position Awareness (PPA), Curriculum Learning (CLIAS), and Spectrum Curvature Limitation (SCL). While the integration of physical principles is intuitive and well-motivated, the reviewers' primary concerns center on:
1. Limited technical novelty, as the proposed methods are applications of well-established ML techniques (auxiliary tasks, curriculum learning, and regularization) rather than introducing new algorithmic contributions
2. Insufficient dataset size (3,170 spectra), which raises strong concerns about the reliability and generalization of deep learning models, particularly Transformers
3. Weak baselines, specifically the lack of comparison with modern graph-based (GNN) or 3D-aware molecular representations
4. Lack of standardized benchmarks and reproducibility, as comparisons with prior work rely on irreproducible splits and unavailable codes, which limits the strength of the empirical validation
5. Imbalanced component contributions, where the reported performance gains appear to be driven primarily by the peak-awareness component (PPA), with CLIAS and SCL providing only marginal improvements.

In addition, there is a clear reference issue that should be addressed. The cited work by Petru Soviany, Radu Tudor Ionescu, Paolo Rota, and Nicu Sebe, titled *“Curriculum Learning in Deep Neural Networks: A literature review,”* appears to be mis-cited. While the author list is correct and the paper title is similar to an existing publication, the venue information is incorrect, as there is no evidence that this work was published in Neural Networks (2022) with the provided DOI. The authors should carefully verify and correct the bibliographic details of this reference to ensure accuracy and proper attribution.

**Reviewer Concerns:**

### Addressed by Rebuttal
- **Dataset Availability**: The authors clearly explain that no larger public dataset exists for full-range (230-400 nm, 1 nm resolution) UV spectra and justify their use of the merged UV-adVISor datasets.
- **Qualitative Analysis**: Additional emphasis on wavelength-wise and qualitative spectral examples helps clarify where improvements occur, particularly in data-sparse regions.

### Still Outstanding
- **Methodological Novelty**: The rebuttal does not sufficiently address concerns that PPA, CLIAS, and SCL are applications of well-known techniques rather than being fundamentally novel methods.
- **Generalization**: The small dataset size remains as a significant limitation. This may cause concerns about overfitting, especially for complex models such as Transformers.
- **Benchmarking**: Comparisons with UV-adVISor remain inherently limited due to irreproducible splits and unavailable code, while no alternative standardized benchmark being provided.
- **Component Contribution**: Among the three proposed components (PPA, CLIAS, and SCL), the empirical gains are dominated by PPA, while CLIAS and SCL contribute only limited additional improvements. This imbalance weakens the claim that all three components provide substantial and comparable contributions.

**Reviewer Scores:**

- **CoJH (2 → 2/3)**: The reviewer acknowledges strong empirical analysis and clear inductive biases, but maintains fundamental objections regarding scope, generality, and the absence of stronger molecular representations. Yet the current rebuttal may not be enough to overturn toward acceptance.
- **L2dy (4 → 4)**: While the rebuttal reiterates empirical improvements and wavelength-specific analysis, fundamental concerns about limited novelty and the small dataset size remain, likely keeping the score unchanged.
- **x34r (4 → 4/5)**: Several technical and presentation issues were clarified in the rebuttal, improving interpretability. However, concerns about methodological novelty and conceptual clarity persist, suggesting at most a marginal score increase.
- **VgMP (2 → 2)**: Core concerns regarding dataset size, lack of standardized benchmarks, and reliance on author-defined splits remain unresolved. The rebuttal failed to address the reasons for rejection from the reviewer.

---

### Decision · Program_Chairs · 2026-01-26

Reject